# Reactive Carbonyl Species and Protein Lipoxidation in Atherogenesis

**DOI:** 10.3390/antiox13020232

**Published:** 2024-02-14

**Authors:** Anne Nègre-Salvayre, Robert Salvayre

**Affiliations:** 1Inserm Unité Mixte de Recherche (UMR), 1297 Toulouse, Centre Hospitalier Universitaire (CHU) Rangueil—BP 84225, 31432 Toulouse CEDEX 4, France; robert.salvayre@inserm.fr; 2Faculty of Medicine, University of Toulouse, 31432 Toulouse, France

**Keywords:** (short chain) RCS, lipoxidation, oxidative stress, protein adducts, inflammation, endothelial dysfunction, atherogenesis

## Abstract

Atherosclerosis is a multifactorial disease of medium and large arteries, characterized by the presence of lipid-rich plaques lining the intima over time. It is the main cause of cardiovascular diseases and death worldwide. Redox imbalance and lipid peroxidation could play key roles in atherosclerosis by promoting a bundle of responses, including endothelial activation, inflammation, and foam cell formation. The oxidation of polyunsaturated fatty acids generates various lipid oxidation products such as reactive carbonyl species (RCS), including 4-hydroxy alkenals, malondialdehyde, and acrolein. RCS covalently bind to nucleophilic groups of nucleic acids, phospholipids, and proteins, modifying their structure and activity and leading to their progressive dysfunction. Protein lipoxidation is the non-enzymatic post-translational modification of proteins by RCS. Low-density lipoprotein (LDL) oxidation and apolipoprotein B (apoB) modification by RCS play a major role in foam cell formation. Moreover, oxidized LDLs are a source of RCS, which form adducts on a huge number of proteins, depending on oxidative stress intensity, the nature of targets, and the availability of detoxifying systems. Many systems are affected by lipoxidation, including extracellular matrix components, membranes, cytoplasmic and cytoskeletal proteins, transcription factors, and other components. The mechanisms involved in lipoxidation-induced vascular dysfunction are not fully elucidated. In this review, we focus on protein lipoxidation during atherogenesis.

## 1. Introduction

Atherosclerosis is a chronic, slow-developing, immuno-inflammatory disease of medium and large arteries and a main cause of cardiovascular morbidity and mortality worldwide [1,2]. Atherosclerosis is characterized by the presence of lipid-rich plaques lining the intima of arteries. These lesions silently and progressively evolve over decades towards plaque instability and rupture, leading to acute vascular events such as myocardial infarction or stroke [2,3,4,5]. Among the mechanisms involved in the pathophysiology of atherosclerosis, reactive oxygen species (ROS) and oxidative stress play a prominent role at each step of the disease [6,7]. A main event in the early stages results from the increased permeability of the endothelium to low-density lipoproteins (LDL), as well as their retention on extracellular matrix (ECM) components and their oxidative modifications, which transform LDLs into highly proatherogenic oxidized LDLs (oxLDLs) [8,9,10]. Lipid peroxidation affects various LDL components, including polyunsaturated fatty acids (PUFAs), cholesterol and other sterols, vitamin E and other antioxidants, and apoB. The uptake of oxLDLs through the scavenger receptor (SR) system leads to the formation of foam cells, which accumulate as fatty streaks—a hallmark of atherosclerosis [11]. 

This review is focused on the effects of lipid peroxidation derivatives of PUFAs, and more specifically on short-chain reactive carbonyl species (RCS) generated from lipoprotein and membrane oxidation [12]. These highly reactive agents include α, β-unsaturated aldehydes (4-hydroxy-2-nonenal or HNE, 4-hydroxy-2-hexenal or HHE, acrolein or ACR), dialdehydes (malondialdehyde or MDA and glyoxal), and ketoaldehydes (methylglyoxal, 4-oxo-2-nonenal or ONE), which are among the most studied so far [13,14]. RCS covalently bind to nucleophilic groups of nucleic acids, phospholipids, peptides, and proteins, to form advanced lipoxidation end products (ALEs) in a process called lipoxidation [12,13,14,15,16]. Recent proteomic studies using mass spectrometry or immunochemical techniques allowed us to identify a huge number of proteins targeted by RCS in atherosclerotic lesions [12,13,14,15,16]. RCS participate in the development of lesions from their initiation during atherogenesis and during their evolution towards more advanced stages. Lipoxidation is associated with a large panel of biological effects, from hormetic and adaptative responses at low physiological levels to alterations of enzymatic properties and signaling dysfunctions in pathophysiological conditions associated with high oxidative stress [17,18,19,20]. 

This review is focused on protein lipoxidation in the vascular wall and its implication in atherogenesis, and to a lesser extent, in advanced plaques. To better understand how and when RCS contribute to these processes, the following section briefly summarizes the different stages of atherosclerosis, with the role of RCS being developed subsequently. 

## 2. Atherosclerosis from the Early Steps to Advanced Lesions: A Brief Overview

### 2.1. Endothelial Dysfunction

Endothelial dysfunction is an early event in atherosclerosis and an independent predictor of future cardiovascular events in patients. A main cause of endothelial dysfunction is the reduced bioavailability of nitric oxide (NO), which plays an essential role in the antihypertensive, antithrombotic, and antiatherogenic properties of the endothelium by regulating blood pressure, vasodilation, and hemostasis [8,9,21]. Various stimuli, including inflammatory circulating agents such as angiotensin II, mechanical forces of shear stress, and cytokines trigger endothelium activation and stimulate the production of ROS, particularly superoxide anion O_2_^•−^. The combination of O_2_^•−^ with NO generates peroxynitrite, which promotes eNOS (endothelial NO synthase) uncoupling and dysfunction [9,22]. Endothelial dysfunction is observed in lesion-prone areas of the arterial vasculature, i.e., arterial bifurcation—bends and curvatures that are exposed to disturbed blood flow, with this being aggravated by risk factors for cardiovascular diseases (CVD) [23,24]. The sensing of shear stress (i.e., intensity, direction, pulsatility) by endothelial cells depends on the presence of shear stress sensors and mechanotransducers, such as the primary cilia, endothelial glycocalyx, PIEZO channels, caveolae, signaling G protein-coupled receptors, protein kinases, and endothelial junctional complexes including PECAM-1, VE-cadherin, and VEGFR2 [24]. Endothelial cell function is modified in response to the disturbed nonlaminar flow, which includes changes in phenotype and gene expression, cytoskeletal rearrangement, the promotion of leukocyte adhesion, mitochondrial stress, and ROS production [25]. CVD risk factors (hypercholesterolemia, hypertension, diabetes) increase endothelial mitochondrial activity, which contributes to endothelial dysfunction through several mechanisms, including mutations in mitochondrial DNA, increased mitochondrial ROS production, and respiratory chain alterations [26]. These events contribute to establishing a “proinflammatory endothelial phenotype,” which implicates the activation of the nuclear factor-kappa-B (NF-κB) transcription factor and the expression of adhesion molecules such as E-selectin and VCAM-1; the procoagulant tissue factor; and the monocyte chemoattractant protein-1 (MCP-1), involved in the recruitment of mononuclear cells in the intima [9,27]. 

### 2.2. LDL Transcytosis 

Endothelial dysfunction increases the permeability of endothelium to macromolecules, which allows the transport of LDLs across endothelial cells to the intima, mainly by transcytosis. This system is independent of the physiological LDL-receptor/clathrin-dependent saturable pathway [28,29,30,31,32,33]. Endothelial LDL transcytosis is a multistep mechanism that includes LDL internalization by endocytosis, vesicular traffic and exocytosis within plasma membrane vesicles e.g. caveolae, which function as shuttles from the apical to the basal cellular side. This transport involves caveolin-1, a protein specific to caveolae, highly expressed in endothelial cells, which regulates mechanical sensing, autophagy, and cellular signaling pathways [34]. Caveolae-dependent transcytosis of LDLs involves the receptor ALK1 (activin-like kinase 1) and the scavenger receptor B1 (SR-B1) [28]. 

### 2.3. LDL Oxidation and Foam Cell Formation

Once in the intima, LDLs are in part retained on proteoglycans and other ECM components and undergo modifications mainly by oxidation. LDL retention and oxidation promote the “environmental response to LDL retention,” which is a critical initiating event in atherosclerosis [35]. It is characterized by an inflammatory response, leading to the recruitment of mononuclear cells in the intima and their differentiation into macrophages. Scavenger receptors promote the uptake of oxidized and modified LDLs by macrophages, their degradation, and the formation of foam cells and fatty streaks [35,36,37]. Within the lesion, inflammatory macrophages secrete cytokines and chemokines and produce high levels of ROS, aggravating the process of LDL oxidation [27,37]. OxLDL contributes to endothelial inflammation via the scavenger receptor LOX-1, the production of ROS, and the activation of inflammatory transcription factors such as NF-κB or the NLRP3 (NOD-like receptor pyrin domain-containing protein 3) inflammasome pathway [38,39,40,41]. The dynamic interplay between monocytes/macrophages and vascular cells, as well as the redox imbalance, maintains a chronic inflammatory environment throughout lesion development [42].

### 2.4. Advanced Lesions

Chemoattractants secreted by macrophages stimulate the migration and proliferation of smooth muscle cells (SMCs) in the intima, which differentiate from the contractile to synthetic phenotype, to form a fibrous cap surrounding fatty streaks [37]. Stable plaques are most often associated with a small necrotic core and a thick fibrous cap less prone to rupture. However, these plaques may develop fibrocalcifications and intraplaque hemorrhages [27,37]. Lesions with thin fibrous caps are unstable and fragile, with a large necrotic core and a cap containing inflammatory cells, macrophages, and a few SMCs [27] (for the definition and classification of advanced atherosclerotic lesions, see [43]). In the intima, SMCs may differentiate into macrophage-like foam cells or osteocytes, leading to intimal calcification [44]. Vascular calcification could be positively (miR-670-3p, miR-3182) or negatively (miR-29b, miR-126-5p) regulated by microRNAs encapsulated into cell-derived extracellular vesicles [45,46,47]. Microcalcifications are the most harmful and are associated with plaque inflammation, which persists throughout advanced lesions with the release of proinflammatory cytokines (IL-1β, IL-6, TNFα) and the risk of plaque rupture [48]. Interestingly, the gut microbiota and its metabolites may impact inflammation, microcalcification, and atherosclerosis outcomes in CVD patients [49]. Likewise, autophagy, which plays an essential role in inhibiting inflammation and promoting efferocytosis in early lesions, could be impaired and even overactivated in advanced vulnerable and unstable plaques, thereby contributing to NLRP3 activation, plaque rupture, and atherothrombosis [50,51]. Efferocytosis inhibition could lead to an accumulation of apoptotic macrophages, the release of matrix-degrading proteases, and increased inflammation [52]. Various forms of cell death affect macrophages and vascular cells, through apoptosis, necrosis, pyroptosis, and ferroptosis (a major cause of endothelial dysfunction and death in atherosclerosis), thereby contributing to plaque destabilization and rupture [53,54]. 

## 3. RCS in Early Atherosclerosis Lesions, from Hormesis to Dysfunctions

RCS generated during LDL oxidation exhibit biphasic properties, from hormetic and protective effects at low levels to dysfunction and toxicity at higher doses. Hormesis is a defense mechanism based on a dose–response relationship by which low levels of stressors upregulate adaptive and protective responses, whereas higher levels become potentially harmful [55]. Low RCS concentrations stimulate hormetic responses by activating signaling pathways such as the transcription factor Nrf2 (nuclear factor erythroid 2-related factor 2 (Nrf2)/Kelch-like ECH-associated protein 1 (Keap1) system), resulting in the expression of cytoprotective and antioxidant enzymes and an enhanced expression of anti-inflammatory cellular defenses [56,57]. Likewise, RCS modulate NF-κB activation or inhibit the NLRP3 transcription factor and its subsequent inflammatory signaling [57]. 

The formation of adducts by RCS on proteins is non-specific, although it is not a totally random process either, as reviewed by Aldini et al. [58]. Indeed, it depends on the local RCS concentration, the expression of neutralization systems, the availability of epitopes susceptible to be modified, and structural factors. Basically, RCS contribute to several steps of atherogenesis, including the formation of foam cells via apoB modification, inflammation via the modification of mitochondrial respiratory chain complexes and antioxidant systems (which enhances ROS production), and the modification of cytoskeleton and ECM proteins, which contribute to endothelial dysfunction. These properties are summarized in Table 1.

### 3.1. LDL Oxidation and Formation of RCS 

LDL oxidation in the vascular wall is a complex mechanism that involves several sources of ROS, including NOXs (NADPH oxidases), the mitochondrial electron transport chain, xanthine oxidase, myeloperoxidase, cellular lipoxygenases, uncoupled eNOS, heme, iron, and copper ions [57,59,60,61,62]. Lipid peroxidation strongly affects polyunsaturated fatty acids (PUFAs) in three steps (initiation, propagation, and termination), with hydrogen abstraction from a carbon and oxygen insertion [63,64]. This peroxidative attack generates a huge variety of lipid peroxidation products, among them lipid peroxyl radicals and lipid hydroperoxides, which undergo structural rearrangements to form RCS [63,64,65,66]. Protein lipoxidation refers to the non-enzymatic post-translational modification of proteins by RCS via their interaction with the nucleophilic side chains of cysteine, histidine, and lysine residues to form Schiff’s bases (addition of the aldehydic group to an amino group of protein), or Michael addition of a nucleophile to α, β-unsaturated aldehydes [18,19]. The formation of RCS adducts is very fast, relatively selective, and depends on the protein microenvironment and the specific epitope exposure. The chemistry of adduct formation on proteins, as well as the reversion mechanisms (conjugation with glutathione catalyzed by glutathione S-transferase, oxidation or reduction by aldehyde dehydrogenase or alcohol dehydrogenase), have been largely described and reviewed [14,18,20,67,68].

Several electrophilic aldehydes are detected in atherosclerotic lesions [12,57]. HNE is one of the most abundant. It is formed by the peroxidation of n-6 PUFAs and could be enzymatically produced by 15-lipoxygenase [20,68]. HNE forms Michael adducts with the highest reactivity for cysteine, followed by histidine and lysine, and the lowest reactivity for arginine [20]. ONE is formed through the oxidation of n-6 PUFAs. It shares structural similarities with HNE, but it is more toxic and reactive on protein nucleophiles, particularly on lysine, on which it forms readily reversible Schiff base that can be oxidized to stable 4-ketoamide [69,70]. 

**Table 1 antioxidants-13-00232-t001:** Main cellular systems modified by RCS.

Systems	Targets	RCS	Epitopes	Consequences	References
**LDL oxidation**	apoB	MDA, HNE	Lys, His, Cys	Foam cells	[66,67]
		ACR	Cys		[71]
**Transcription factors in inflammation**					
NF-kB	IkBα	HNE	Cys	Inhibition	[72,73]
	IKK	HNE, ACR	Cys-179	id.	[74,75,76,77,78]
NLRP3		HNE	Cys	Inhibition	[79]
Nrf2	Keap1	HNE, RCS	Cys	Antioxidant	[80]
**Mitochondria**	Complex I	HNE, ONE		Decreased activity	[81]
	Complex II	ACR		ROS increase	[82,83,84,85,86,87,88,89,90]
	ANT	HHE, HNE		Apoptosis	[91,92]
**Antioxidant systems**					
ALDH2		HNE, ONE		ROS increase	[93]
GSTA4		HNE		id.	[94]
TRX-1		HNE, ACR	Cys73	id.	[95]
PRX6	HNE, ONE		Cys-91, Lys-209	Inhibition	[96]
**eNOS**	ONE		Lys	Decreased activity	[97]
	GTPCH	HNE		id.	[98]
**Endothelial barrier components**					
Glycocalyx	HPSE	ACR, MDA	Lys	Degradation	[99]
Cytoskeleton	Actin	HNE	Cys-374	Stress fibers	[100,101]
	Vimentin	RCS	Cys-328	Altered motility	[102,103]
**Growth factor receptors**					
	PDGFR,	HNE, ACR		Altered signaling	[104,105,106,107,108]
	EGFR				-
**Endothelial senescence**					
TXNIP	PPARγ	HNE	His-413	Senescence	[109]
SIRT1		HNE, ONE	Cys	id.	[110]
20S proteasome		HNE		Inhibition, ROS	[111,112,113,114,115]
**ER stress**	GRP78	HNE, ONE	His, Lys	Apoptosis,	[116,117]
	PDI HNE		Cys	Apoptosis	[118]
**Cell death**	CDR	HNE		Apoptosis	[119]
	ANT	HHE		Apoptosis	[92]
	VDAC2	HNE, ONE	Cys-210	Ferroptosis	[120]

ACR, acrolein; ALDH, aldehyde deshydrogenase; ANT, adenine nucleotide translocator; CDR, cell death receptor; Cys, cysteine; GSTA4, glutathione S-transferase A4; GTPCH, GTP cyclohydrolase; HHE, hydroxyhexenal; His, histidine; HNE, hydroxynonenal; Lys, lysine; ONE, oxononenal; PRX, peroxiredoxin; Trx-1, thioredoxin-1; TXNIP, thioredoxin interacting protein; VDAC2, voltage-dependent anion-selective channel protein 2.

Acrolein (ACR) is an environmental volatile pollutant present in tobacco smoke and in cooking and exhaust fumes. It is endogenously formed by the peroxidation of PUFAs and through the metabolism of amino acids and polyamines. Acrolein rapidly reacts with cysteine, histidine, and lysine and is detected in oxLDLs and human atherosclerotic lesions [12]. MDA is a product of PUFA peroxidation, abundantly present in oxLDLs as MDA-lysine adduct of apoB. MDA is highly mutagenic, cytotoxic and carcinogenic. It is largely used as a biomarker of lipid peroxidation to evaluate the extent of oxidative stress [121,122]. RCS formation and structure are shown in Figure 1.

### 3.2. Protein Lipoxidation in the Vascular Wall

#### 3.2.1. Modification of apoB by RCS in LDL: A Main Role in Foam Cell Formation

A variety of oxLDLs could be detected in the intima, from minimally/mildly oxLDLs mainly oxidized on their lipid moiety to heavily oxidized LDLs with RCS-modified apoB. Mildly oxLDLs are taken up through scavenger receptors (SR) such as LOX-1, present on endothelial cells or CD36, which is expressed by SMCs, macrophages, and endothelial cells. These mildly oxLDLs are highly inflammatory and contribute to endothelial dysfunction [123,124,125]. Extensively oxidized LDLs contain large amounts of oxidized lipids, with apoB being modified by RCS (MDA, HNE, ACR), which deviates their uptake and metabolism towards the scavenger receptor class A (SR-A) pathway in macrophages [126,127]. MDA- or HNE-modified LDLs are a main cause of foam cell and fatty streak formation [128,129,130]. MDA specifically reacts with the terminal ε-amino group of lysine residues involved in the recognition of LDLs by the LDL receptor [121,122]. As described by Lankin et al. [122], MDA-LDLs undergo changes in the molecular conformation of apoB, which promotes the formation of cross-links between LDL particles and changes in electrophoretic patterns pointing out larger LDL formations. 

HNE is more effective than MDA for modifying and increasing LDL negative charge and global molecular weight [68]. HNE-modified LDLs are taken up by macrophages and generate foam cells [14,129,130]. In HNE-modified apoB, Lys residues are the main target of HNE, and the other modified amino acid residues are tyrosine, serine, histidine, and cysteine [130]. In ACR-modified LDLs, ACR–apoB adducts are mainly formed on Lys residues, which promotes their rapid uptake by macrophages through SR-A1 [71].

LDL oxidation and some consequences of RCS release on protein lipoxidation possibly occurring in the intima are summarized in Figure 2.

The presence of RCS adducts in lesions in human and animal models for atherosclerosis [131,132,133,134] was demonstrated by immunocytochemistry and immunofluorescence techniques using specific anti-HNE and anti-MDA antibodies. Other adducts could be detected including Michael addition-type 4-hydroxy-2-hexenal (HHE)–histidine adducts [132], or ACR–lysine adducts (N-alpha-acetyl-N-epsilon-(3-formyl-3,4-dehydropiperidino) lysine [133]. LDLs and lipid peroxidation within the subendothelial area have multiple consequences—not only the accumulation of foam cells and fatty streaks but also the formation of RCS adducts on cellular and extracellular proteins. RCS adducts are detected on cellular membranes and ECM protein components of atherosclerotic lesions [12,126], suggesting their possible implication in atherogenesis. 

#### 3.2.2. RCS and Inflammation

##### Lipoxidation of Transcription Factors and Inflammation 

Inflammation is an early component of plaque [27]. In the “response to retention” hypothesis [135], it was proposed that immuno-inflammation in early lesions is a defense mechanism tending to counter the accumulation of modified and oxidized LDLs by recruiting mononuclear cells, which may remove harmful oxLDLs from the environment. In contrast, the chronic inflammatory condition of advanced lesions is a key factor of plaque development and instability. Low RCS concentrations stimulate the activation of the redox-sensitive transcription factor NF-κB and concomitantly activate the anti-inflammatory nuclear factor (erythroid-derived 2)-like 2 (Nrf2) [136,137,138]. High RCS levels elicit an anti-inflammatory response via at least two mechanisms: 1) inhibiting inflammatory transcription factors (NF-κB and the NLRP3 inflammasome pathway) [139] and 2) stimulating the expression of endogenous antioxidant defenses via Nrf2 [137]. 


*NF-κB*


NF-κB is a key regulator of inflammation and cell survival evoked by proatherogenic stressors [136,138]. In unstimulated cells, NF-κB is sequestered in the cytoplasm in an inactive state by its inhibitor, IκBα [136]. Upon stimulation (by oxidative stress), IκBα is phosphorylated by IκB kinase (IKK), a redox-sensitive regulator of NF-κB activation. This promotes IκBα degradation by the ubiquitin/proteasome pathway and the translocation of NF-κB into the nucleus where it binds specific DNA domains and induces the expression of inflammatory genes [136,138]. This includes cytokines, chemokines, macrophage chemotactic factor (MCP)-1, matrix metalloproteinases (MMPs), cyclo-oxygenase (COX)-2, inducible nitric oxide synthase (iNOS), vascular endothelial growth factor (VEGF), adhesion molecules (VCAM-1, ICAM-1, E-selectin) [138]. 

At low levels (lower than 10 µM), HNE stimulates the phosphorylation of IκBα and the binding of NF-κB to DNA, which induces MMP2 expression and SMCs proliferation [140,141]. In macrophages, HNE activates NF-κB via the EGFR/p38 MAPK pathway, thereby promoting the expression of 5-lipoxygenase and the generation of leukotrienes [142]. In human U937, HNE and 27-hydroxycholesterol trigger an inflammatory response via Toll-like receptor 4 (TLR4) and NF-κB, leading to cytokine release and MMP-9 upregulation [143]. Likewise, hydroxyhexenal (HHE)-induced NF-κB activation upregulated p38 MAPK and ERK activities in endothelial cells [144]. At higher concentrations (10 µM and higher), HNE prevents the activation of NF-κB by LPS in human monocytes by inhibiting the phosphorylation and proteasomal degradation of IκBα [72]. This inhibitory mechanism could result from the formation of HNE adducts on IκBα, possibly leading to a modification of protein conformation, preventing its phosphorylation by IKK, as reported in hepatocytes in a murine model of alcoholic liver disease [73]. HNE forms adducts on IKK, particularly on cysteine-179 [74,75], which inhibits IKK signaling and NF-κB activation [76]. Similar observations were reported for ACR, which modifies IKK [77], thereby inhibiting IκBα phosphorylation and NF-κB activation [78]. The inhibitory effect of RCS on NF-κB blocks the expression of the inducible NO synthase (iNOS) (a target gene of NF-κB) and NO production evoked by LPS and interferon-γ in SMCs [145]. In addition, high HNE levels modify c-Jun NH2-terminal kinase (JNK) and upregulate the activating protein-1 transcription factor (AP-1), which promotes apoptosis [146].


*NLRP3*


NLRP3 is a major proinflammatory protein complex when associated with the adaptor ASC protein and caspase 1. It plays an important role in atherogenesis [41,139,147]. It is activated by oxLDLs, ROS, cholesterol crystals, and other danger signals, leading to the release of the inflammatory cytokine IL-1β, which aggravates inflammation and promotes cell death by pyroptosis [148]. Recently, Hsu et al. [79] reported an inhibitory effect of HNE on NLRP3 activation via a direct binding of HNE to NLRP3 cysteines. This modification alters the interaction between NLRP3 and NEK7, which is an essential partner of inflammasome assembly and activation [149]. This mechanism could be reversed by N-acetylcysteine and GSH [149].


*Nrf2*


Nrf2 is a main regulator of cellular resistance to oxidative stress and electrophiles and a major protective system in atherosclerosis [56,150,151,152,153]. This transcription factor controls the expression of antioxidant/detoxifying genes and proteins, which prevents and protects against the onset of oxidative stress outcomes [56,150,151]. In basal conditions, Nrf2 is associated with Keap1 in the cytoplasm. The phosphorylation of Keap1 by GSK-3β promotes its proteasomal degradation after ubiquitination [150,151,152]. Oxidative stress stimulates the release of Nrf2 from the complex with Keap1, allowing its translocation into the nucleus. Nrf2 binds to antioxidant response elements (AREs) on DNA, which initiates the expression of antioxidant and protective genes, including NAD(P)H quinone oxidoreductase 1, glutamate-cysteine ligase, sulfiredoxin 1 and thioredoxin reductase 1, heme oxygenase-1 (HO-1), glutathione S-transferase, the cystine/glutamate amino acid transporter, and other protective systems [56,150,151,152,153]. The Nrf2/Keap1 signaling pathway is highly sensitive to electrophiles [80], which stimulate the expression and nuclear translocation of Nrf2, providing an adaptive response to cellular stress. The mechanisms by which RCS activate Nrf2 involve the presence of several cysteine residues in Keap1, which are highly susceptible to modification by electrophiles [80], leading to Keap1 degradation and Nrf2 nuclear translocation. Nrf2 is a main effector of the hormetic responses evoked by low RCS levels in vascular cells [56,146,151], via the upregulation of antioxidant, cytoprotective, and antiapoptotic systems, including HO-1 and peroxiredoxin-1. Nrf2 regulates proteasome and autophagy activities [146,154] and stimulates GSH synthesis, which prevents the modification of proteins by RCS [118]. 

Despite its antioxidant and cytoprotective properties, the role of Nrf2 in atherosclerosis is debated [155]. Nrf2 activation stimulates the expression of the scavenger receptor CD36 and the formation of foam cells [156] and could increase the expression of proinflammatory genes in more advanced stages of atherosclerosis [155]. As reported by Harada et al. [157], Nrf2 inhibition could be atheroprotective in advanced plaques. All in all, Nrf2 may exhibit pro- or anti-atherogenic properties, depending on the (early or advanced) development of atherosclerotic lesions.

##### Cyclooxygenase-2 Activation

Cyclooxygenase-2 (COX-2) is a key enzyme involved in the production (from arachidonic acid) of high prostaglandin levels during inflammation and immune responses, particularly in vascular pathophysiology [158]. COX-2 is rapidly induced in response to ROS, cytokines, growth factors, and HNE, which stimulates COX-2 expression in part via the activation of the p38 MAPK pathway [158]. HNE-induced COX-2 could result from an accumulation of p53 and sp1 transcription factors (and ubiquitinated proteins) due to a downregulation of proteasome [158]. Other RCS such as ACR or ONE are unable to stimulate COX-2 expression, which could be HNE-specific. The lipoxidation mechanisms possibly involved in COX-2 expression by HNE are not yet elucidated.

##### RCS and ROS 

GSH depletion by RCS generates ROS and redox imbalance [75]. Other mechanisms involve the formation of RCS adducts on antioxidant enzymes or alterations of eNOS activity. RCS may also inhibit ROS production, as observed in neutrophils in which HNE could modify and inactivate proteins involved in the respiratory burst (ROS production) and phagocytosis, which reduces both inflammation and antimicrobial defenses [159]. 


*Effect of RCS on Mitochondrial ROS Production*


The mitochondrial electron transport chain is an important source of endogenous cellular ROS [81], and it is also a main target for RCS [160,161,162]. HNE, MDA or ACR affect respiratory chain activity, decrease the mitochondrial membrane potential, and generate mitochondrial ROS [82,83,84,85]. HNE exogenously added or endogenously produced in mitochondria via cardiolipin oxidation [86], decreases the activity of mitochondrial complexes -I and -II [87,88]. Several mitochondrial proteins could be modified by HNE or ONE, such as the FAD-containing subunit of succinate dehydrogenase, an essential component of succinate: ubiquinone oxidoreductase (or mitochondrial complex II) [89]. Hwang et al. [90] identified several mitochondrial proteins modified by HNE in cardiomyocytes during diabetes, among them NADH dehydrogenase (ubiquinone), iron–sulfur protein 3, aconitate hydratase-1, and heme proteins (myoglobin and cytochrome c1), along with the decreased activity of mitochondrial respiratory chain complex proteins. HNE and HHE form adducts on UCPs and adenine nucleotide translocase (ANT), which contributes to mitochondrial uncoupling by increasing proton leak, regulating membrane potential, and triggering mitochondrial dysfunction [81]. Increased mitochondrial ROS and dysfunction are involved in HNE-induced vascular SMC apoptosis [91]. 

On the other hand, low HNE concentrations may limit ROS production in mitochondria via the activation of the proton transporter function, leading to mild uncoupling that decreases the production of mitochondrial O_2_^•−^ [163]. This mechanism, associated with the modulation of redox-regulating enzymes in mitochondria, could be involved in the activation of the Nrf2/ARE signaling by HNE [163].


*Effect of RCS on Antioxidant Systems*


-
*ALDH2*


Mitochondrial aldehyde dehydrogenase 2 (ALDH-2) is an oxidizing enzyme present in mitochondria and involved in the detoxification of RCS. ALDH-2 could act as a defense mechanism against oxidative stress in cardiovascular diseases. At low levels, HNE and ONE are degraded by ALDH-2, whereas at higher levels, these agents form covalent modifications on this enzyme and inhibit its activity [93]. 

-
*Glutathione-S Transferases (GSTs)*


The conjugation of aldehydes with GSH is a major detoxifying mechanism of reactive electrophiles, which prevents their reaction with cellular nucleophiles and facilitates their elimination. The conjugation with GSH may spontaneously occur, but it is facilitated by enzymes such as the cytosolic glutathione transferases (GSTs) which promote the reduction of hydroperoxides to form oxidized glutathione (GSSG) [94,164]. GSTs protect against oxidative injury and regulate GSH homeostasis [165]. The GSTA4-4 isoform is the most selective for catalyzing the conjugation of GSH with RCS and is a main defense mechanism against oxidative stress [81,91,166]. However, the formation of HNE adducts on the catalytic site of GST inhibits its activity and promotes oxidative stress [166]. 

-
*Thioredoxin 1*


Thioredoxin (Trx-1) is a key antioxidant enzyme involved against oxidative stress through its disulfide reductase activity regulating protein dithiol/disulfide balance. ACR and HNE react with Trx-1 on cysteine-73, inhibiting its enzymatic activity, which potentiates ROS production and promotes monocyte adhesion to endothelial cells [95].

-
*Peroxiredoxins*


Peroxiredoxins (PRXs) are ubiquitous peroxide and peroxynitrite-scavenging enzymes [167]. The modification of PRX1 and PR6 by RCS has been reported [96,168]. PRX6 is an important antioxidant protein present in various tissues including cardiac muscle, skin and lung. HNE and ONE promote PRX6 modification and the formation of cross-links, particularly the formation of adducts on cysteine-91-lysine-209, which induces conformational changes and protein inactivation [96].


*Effect of RCS on eNOS*


The modification of eNOS by HNE and ONE was reported in preeclamptic placentas and in cultured trophoblasts, with subsequent decreased NO generation and trophoblast migration [97]. Proteomic studies of recombinant eNOS modified by ONE showed the modification of several lysine residues on both oxidase and reductase domains, inhibiting its enzymatic activity [97]. So far, a direct modification of eNOS in the vascular wall is not known. However, HNE could promote eNOS uncoupling and O_2_^•−^. generation via a depletion in tetrahydrobiopterin (BH4, eNOS co-factor), resulting from the modification by HNE and subsequent proteosomal degradation of the GTP cyclohydrolase I (GTPCH), that is involved in BH4 biosynthesis [98]. Another mechanism could implicate an inactivation of Akt by HNE [169,170], the phosphorylation of eNOS by Akt on serine-1179 being required for its activation and NO production [169].

#### 3.2.3. Lipoxidation of Endothelial Barrier Components 

In physiological conditions, endothelial cells form a semi-permeable barrier to blood constituents, i.e., cells, macromolecules, albumin, and bioreactive agents. During atherogenesis, alterations of endothelial cell barrier integrity contribute to endothelial dysfunction and increased permeability to LDLs. RCS generated and released during the LDL oxidation process in the intima may play a role in endothelium dysfunction.

##### Glycocalyx

The glycocalyx is an extracellular matrix component surrounding the endothelium, as an interface between the vascular wall and circulating blood. Endothelial glycocalyx consists of glycoproteins, proteoglycans, glycosaminoglycans, hyaluronic acid, and associated plasma proteins. It is secreted by endothelial cells and located on the luminal side of vessels [171,172]. Glycocalyx contributes to mechanotransduction signals in response to stimuli and shear stress and maintains vascular permeability barrier and NO release [173,174]. The degradation of glycocalyx components, particularly heparan sulfate, by heparanase (HPSE), contributes to increased endothelial cell permeability, LDL retention, SMC migration, and intimal ECM remodeling [175]. Exposure of heparan sulfate to MDA or ACR promotes its degradation [99,122]. ACR modification on lysine residues of the inactive proform of heparanase (proHPSE) triggers its activation and heparan sulfate degradation, which increases endothelial cell permeability [99].

##### Extracellular Matrix Proteins

Endothelial cells are anchored on an underlying basement membrane, the intimal ECM, containing several components including laminin, collagens, fibronectin, heparan sulfate, proteoglycan, or perlecan [176]. The interaction of endothelial cells with the basement membrane maintains the integrity of the vascular wall [177]. ECM in the media has a more specialized structure, with elastin/fibrillin/fibulins/microfibril glycoprotein-associated matrices, as well as various components, including type IV collagen, laminins, perlecan, nidogens, or fibronectin. The medial ECM maintains the phenotype and function of contractile SMCs [176,177,178]. Atherogenic inducers and proinflammatory agents, leukocyte and monocyte infiltration, or foam cell accumulation promote ECM remodeling, as well as SMC migration and proliferation in the intima [176,177,178].

RCS released by oxLDLs on the connective tissue in the intima could bind and modify ECM protein components, such as collagen, laminin, or fibronectin. OxLDLs promote in vitro the formation of MDA adducts on fibronectin, laminin, and collagens type I > type V and type III > type IV > type II [179,180]. MDA-modified fibronectin is detected in human atherosclerotic lesions, and antibodies specific to this MDA-modified fibronectin could be correlated with the extent of CVD [180]. Likewise, the presence of MDA-modified laminin was detected in human atherosclerotic lesions and apoE-KO mice, associated with the induction of anti-MDA-modified laminin antibodies, and correlated with a more aggressive development of atherosclerosis [181]. Duner’s group also identified the presence of MDA-collagen type IV in human endarterectomy lesions. In vitro MDA-modified collagen type IV altered the attachment of endothelial cells and stimulated the expression of VCAM-1 adhesion molecule, suggesting an implication of collagen modification in endothelial dysfunction [182].

HNE–histidine adducts are age-dependently detected in the three layers of the arterial wall, with a strong expression in the intima associated with atherosclerotic lesions in the media and the adventitia [183]. These findings confirm that HNE is a main marker of vascular oxidative stress and lipid oxidation and suggest a role for this aldehyde in the development of vasa vasorum and microcapillaries. However, HNE and other RCS do not modify vascular elastin [183], in contrast to elastin in skins exposed to UV radiations [184]. In vitro, HNE inhibits the elastogenic activity of TGFβ by forming adducts on EGF receptor, which activates a downstream signal inhibiting TGFβ-induced responses [185], in agreement with the inhibitory signaling of EGF on tropoelastin expression by TGFβ [186].

##### Cytoskeleton Proteins

Cytoskeletal actin, intermediate filaments and microtubules; focal adhesion kinases (FAKs); and adherens junction proteins involved in the regulation of the endothelial barrier [187] are targeted by RCS. 


*Actin*


RCS are potent inducers of actin stress fibers and actin aggregation, via mechanisms implicating ERK1/2, p38 MAPK, JNK, and redox imbalance [188,189]. The pretreatment of actin with HNE alters the structure of actin filaments [100,101]. LC-ESI-MS/MS studies allowed identifying the site of actin modification by Michael addition of HNE to cysteine-374 [100,101].


*Tubulin*


RCS (HNE and ONE) form adducts on purified bovine brain tubulin, resulting in lysine-dependent protein cross-linking and inhibition of tubulin polymerization, with ONE being more potent than HNE as cross-linker and inhibitor of tubulin assembly [190]. In addition, LC-MS/MS analysis demonstrated the modifications of several cysteine residues by HNE. In vitro, HNE triggered the destruction of the microtubule network in fibroblasts and in the PC12 cell line (used as model for neuronal differentiation) [191]. This microtubule disruption by HNE is well described in neurons, possibly via the formation of cysteine adducts on tubulin [192,193]. Though not yet reported in the vascular wall, an impairment of tubulin function and cytoskeletal alterations evoked by RCS could be hypothesized in the chronic oxidative stress context of atherosclerosis.


*Vimentin*


Vimentin is the main protein of endothelial type III intermediate filaments and an important factor of stability and tissue integrity, against mechanical forces exerted by the blood flow [194]. Links between vimentin, microfilaments, and microtubules coordinate cell polarization and migration as well as endothelial cell function, inflammation, and atherogenesis [194]. Intermediate filaments (vimentin and lamin) are highly sensitive to oxidative and electrophilic stress, which promote their disruption and fragmentation, leading to the formation of aggresome structures [102]. HNE forms adducts on nucleophilic (cysteine, histidine, lysine) residues on vimentin, with cysteine-328 being a main target. Cysteine-328 may act as a hub for electrophilic modifications, leading to intermediate filament rearrangements and an extensive reorganization of the vimentin cytoskeletal network, possibly acting either as a mechanism of defense or as a mediator of cell damages [102,103]. Vimentin lipoxidation alters the motility and the contractile capacity of fibroblasts [18]. Recent reports indicated that vimentin in macrophages could play a role in CD36 trafficking and foam cell formation [195]. From these observations, it could be interesting to evaluate the consequences of vimentin modification by RCS on the accumulation of foam cells in atherosclerotic lesions.


*Integrins and focal adhesions*


Interactions between integrins, focal adhesion, and cytoskeleton maintain vascular permeability, cell–matrix adhesion, and cell shape. Any alteration of these links results in a disruption of EC barrier function and intercellular gap formation [189]. HNE could form adducts with focal adhesion, adherens junction proteins or α5 and β3 integrins, thereby modifying their structure and impairing the endothelial barrier function [189].

##### Cell Signaling Kinases and Growth Factor Receptors

In early atherosclerotic lesions, vascular cells and macrophages release various inflammatory cytokines, lipid mediators, and growth factors implicated in the formation of the fibrous cap. SMC phenotypic changes from a contractile to a synthetic state are associated with SMC migration and proliferation in the intima [4,196,197,198]. Several signaling pathways are modulated by RCS, including MAP kinases, PKC isoforms, cell-cycle regulators, receptor tyrosine kinases, and caspases. The formation of HNE or ACR adducts on receptor tyrosine kinases (RTKs) (PDGFR and EGFR) has been detected either in vitro in vascular cells or in vivo in human carotid endarterectomy plaque, in atheroclerotic lesions of hypercholesterolemic rabbits and apoE^−/−^ mice [57,104,185]. At low concentrations, these modifications activate the downstream signaling cascade of RTKs, including src, PI3K/Akt, and ERK1/2, leading to cell survival and proliferation [105,106,107,108,169,199,200]. High HNE levels extensively modify RTKs and trigger their progressive dysfunction. For instance, the accumulation of HNE adducts on PDGFR limits its affinity for PDGF, which decreases PDGF-stimulated cell proliferation and migration [57]. Likewise, the accumulation of HNE adducts on EGFR inhibits the PI3K/Akt pathway and promotes a switch toward apoptosis [108].

### 3.3. Lipoxidation in Advanced Lesions 

#### 3.3.1. Endothelial Senescence

Cellular senescence is a main feature of endothelial dysfunction characterized by the proinflammatory and prothrombotic phenotype of endothelial cells [201,202,203]. Senescent endothelial cells are detected in human atherosclerotic lesions [202,203]. Several features characterize senescence, including the progressive shortening of telomeres, growth arrest, increased expression of the cyclin-dependent kinase inhibitors p21 and p16, increased cell size, increased cytoplasmic activity of senescence-associated β-galactosidase (SA-β-Gal), decreased sirtuin activity, and loss of proteostasis [204,205,206]. Endothelial senescence is a cause of dysfunction with consequences in vascular remodeling, angiogenesis, and secretion of inflammatory factors [205,206]. RCS (HNE, ONE, acrolein) are known triggers of age-related signaling pathways [207]. Riahi et al. [109] reported that HNE secreted by foam cells could promote endothelial senescence via an increased expression of the pro-oxidant thioredoxin-interacting protein (TXNIP), resulting from an activation by HNE of the peroxisome proliferator-activated receptor (PPAR)δ. The mechanism of PPARδ activation could result from the high binding affinity of HNE for histidine-413 in the ligand-binding domain in PPARδ, comparable to that exerted by eicosapentanoic acid [208,209]. 


*Sirtuins*


Silent information regulator proteins (sirtuins or SIRTs) are a family of nicotinamide adenine dinucleotide (NAD)-dependent histone deacetylases involved in the deacetylation of histones and non-histone proteins [210]. SIRT1 protects endothelial cells from oxidative stress, inflammation, and senescence, while its overexpression prevents endothelial dysfunction and atherogenesis [211]. SIRT1 is activated by mild oxidative stress and lipid peroxidation, and it is inhibited by chronic inflammation in advanced lesions [212]. RCS (HNE, ONE) form adducts on SIRT1 on cysteine residues, thereby promoting its degradation and the accumulation of acetylated proteins characteristic of “inflammaging” and a hallmark of senescence [110]. 


*Proteasome and autophagy*


RCS stimulate several aging-related signaling pathways, including cyclin-dependent kinase (CDK) inhibitors p16 and p21, which could be mobilized in response to aldehyde-induced DNA damages as mediators of cell cycle arrest [207]. Likewise, HNE and ONE alter proteasome activity, which declines with aging, resulting in an accumulation of undegraded ubiquitinated material [57,207]. During early atherogenesis, low RCS concentrations stimulate proteasome activity, which, in turn, activates Nrf2 and the expression of antioxidant systems. In contrast, in advanced lesions, the accumulation of oxidized substrates, aggregates, and cross-linked proteins tends to inhibit proteasome activity [111,112,113]. The formation of HNE adducts on the chymotrypsin-like activity of the 20S proteasome sub-unit modifies its catalytic site and inactivates its enzymatic activity [114]. HNE forms an unstable adduct on the α7 subunit of the 20S proteasome, resulting in decreased proteasomal activity and ROS generation [115]. 

Autophagy triggers the degradation of proteins or cellular organelles engulfed in a double-membrane vacuole, the autophagosome, which becomes an autophagolysosome after fusion with lysosomes [213]. In advanced lesions, autophagy could be activated in response to inflammation, oxLDLs, or RCS, to maintain tissue homeostasis [214]. High RCS levels impair the formation of autophagosomes, thereby promoting ferroptosis [215]. In cultured endothelial cells, ACR-induced autophagy alters lysosomes and mitochondria, leading to apoptosis [216]. The mechanism by which RCS activate autophagy is not elucidated and does not involve a direct modification of autophagy components. 

#### 3.3.2. Lipoxidation and ER Stress

A consequence of altered protein homeostasis evoked by oxidative stress and RCS reactivity is the activation of ER stress and the unfolded protein response (UPR), together with autophagy [217,218,219]. ER stress is activated as a survival mechanism, allowing cells to recover from damages in response to pathological factors, leading to the accumulation of unfolded or misfolded proteins in the ER lumen [218,219]. In the case of prolonged and intense stress, ER stress switches to apoptosis [219]. 

HNE, ACR, and MDA modify various ER components, particularly ER chaperones, including heat shock protein 70 (Hsp70), Hsp90, protein disulfide isomerase (PDI), and GRP78 [118,155,220]. The modification of Hsp70 by HNE promotes its cleavage by calpain and apoptosis [221]. GRP78 is involved in the control quality function of ER and maintains the inactivity of ER sensors IRE-1α, PERK, and ATF6 at the ER membrane [222]. HNE and ONE form adducts on lysine and histidine residues on GRP78 near the ATPase domain, which inhibits its activity and promotes apoptosis [116,117].

#### 3.3.3. Lipoxidation and Cell Death 

Vascular cell death is a major event present at each step of atherosclerosis, from atherogenesis to advanced lesions and plaque rupture [223]. Inflammatory factors, oxLDLs, lipid peroxides, and RCS promote several cell death signaling mechanisms by apoptosis, necrosis, necroptosis, pyroptosis, or ferroptosis [53,54,221,223,224,225]. The accumulation of RCS adducts on proteins triggers the dysfunction of vascular homeostasis, ER stress [226], and proteasome inhibition, leading to cell death [154,221]. Apoptosis and ferroptosis are among the most investigated cell death events evoked by RCS.


*Apoptosis*


Apoptotic signaling in vascular cells involves both intrinsic mitochondrial apoptotic pathways and extrinsic cell surface death receptors [227]. The extrinsic pathway includes the TNF receptor, Fas/CD95, or TRAIL (TNFα-related apoptosis-inducing ligand). Upon activation, these receptors recruit adapter molecules to form a death-inducing signaling complex (DISC), which promotes the binding of caspase 8 and the activation of a downstream apoptotic signaling mechanism [228]. HNE activates Fas-ligand and TRAIL apoptotic signaling via unelucidated mechanisms, an hypothesis being the modification (by HNE) of ligand-binding sites on the death receptors [119,229]. Likewise, ACR potentiates TRAIL-induced apoptosis by upregulating the expression of death receptors and downregulating Bcl2 [230]. 

RCS contribute to the intrinsic mitochondrial apoptotic pathway, which mainly results from excessive ROS production by mitochondria, leading to mitochondrial dysfunction, loss of membrane potential, and disruption of cytosolic calcium homeostasis [224,227]. These events promote the opening of the membrane permeability transition pore (MPTP), causing mitochondrial swelling, cytochrome C release, and apoptotic or necrotic cell death [224,227]. HNE contributes to this apoptotic pathway by stimulating JNK activation via ER stress and mitochondrial ROS production via GSH depletion and cardiolipin oxidation [119,229]. A modification of the adenine nucleotide translocator (ANT) by HHE was reported, leading to MPTP opening and apoptosis [92]. 


*Ferroptosis*


Ferroptosis is a form of non-apoptotic cell death depending on iron-mediated lipid peroxidation, a decrease in glutathione peroxidase-4 (GPX4) activity and GSH content, and the disruption of mitochondrial structure, with major implications in endothelial cells and vascular diseases [54,231]. As reported by Chen et al. [120], the inhibition of aldehyde deshydrogenase-1 (ALDH1) resulted in ferroptosis associated with an accumulation of HNE and ONE adducts on several targets, particularly on cysteine-210 in voltage-dependent anion-selective channel protein 2 (VDAC2). VDAC2 is a pore-forming protein present at the outer membrane of mitochondria, which could prevent apoptosis by interacting with the proapoptotic protein BAK [232]. Interestingly, VDAC2 was reported to interact with and bind eNOS in pulmonary endothelial cells, stimulating, in turn, NO production [233]. One could hypothesize that VDAC2 modification by HNE and ONE may occur in vascular endothelial cells, which could locally decrease NO production and trigger endothelial ferroptosis. Of note, Chen’s group, using a quantitative chemoproteomic method to profile protein carbonylation, identified more than 400 carbonylated proteins including lipoxidation-modified molecules possibly involved in ferroptosis [120]. Amoscato et al. [234] recently reported the formation of protein adducts with hydroperoxy-phosphatidylethanolamine (PE) electrophilic cleavage products during ferroptosis and identified several PE-lipoxidated proteins possibly involved in ferroptosis. The precise mechanism by which protein lipoxidation promotes ferroptosis is not yet elucidated [235].

#### 3.3.4. Lipoxidation and Angiogenesis

Intraplaque angiogenesis mainly develops in advanced lesions, in which it may lead to bad outcomes, including macrophage infiltration, inflammation, and intraplaque hemorrhage, which is a main cause of plaque rupture and thrombosis [236]. Low levels of oxLDL and RCS (HNE, ACR) promote the formation of neocapillaries within the plaque via the expression and secretion of proangiogenic factors such as VEGF or sphingosine-1 phosphate [236,237]. In human endarterectomy lesions, HNE adducts colocalize with CD31 (a marker of endothelial cells), suggesting a close relationship between HNE and neovessel formation [238]. At higher concentrations, HNE (and oxLDL) inhibits angiogenesis and could promote endothelial cell death, thereby increasing the risk of intraplaque hemorrhage and rupture [236,238].

#### 3.3.5. Lipoxidation and Vascular Calcifications

Vascular calcification in advanced lesions is considered an aggravating event associated with predictable cardiovascular morbidity. Inflammation is involved in the formation of spotty or granular calcifications (“microcalcifications”), which could be associated with the M1 (inflammatory) macrophage phenotype and plaque rupture [239,240]. By contrast, the transdifferentiation of vascular SMCs into osteoblast-like cells could promote sheet-like calcification or “macrocalcification,” which is more protective and could stabilize the plaque [239,240]. The molecular mechanisms involved in micro- or macrocalcification are still unclear. However, there is some evidence that oxidized lipids and ALEs (advanced lipoxidation end products) could promote vascular microcalcifications through yet-unsolved mechanisms [239,240]. Of note, HNE adducts and oxLDLs were detected around calcium deposits in stenotic aortic valves, suggesting a role for RCS in the calcification process [241]. 

## 4. Pharmacological Interventions for Preventing and Neutralizing Lipoxidation

Nutritional interventions could affect lipoxidation and its derived molecular and cellular damages by modifying membrane fatty acid unsaturation, or via caloric restriction. Several studies summarized by Zadeh et al. [121] showed that such approaches were associated with a reduction in MDA adduct deposits in a variety of tissues, mainly in rodents.

Effective therapeutic studies aimed at limiting or neutralizing the formation of RCS and RCS adducts on proteins should theoretically prevent the vascular complications of atherosclerosis. Several pharmacological or dietary approaches to counteract lipid oxidation and its consequences in the vascular wall have been developed for years [6,11,59]. However, most therapeutic antioxidant interventions were ineffective or underperforming in their capacity to inhibit the occurrence of cardiovascular events in clinical trials [3]. In this article, we briefly summarize some approaches able to prevent or reduce RCS adduct formation and lipoxidation (reviewed in ref. [6]).

Dietary antioxidants and metal chelators block ROS production and PUFA oxidation. As reported by Salekeen et al. [6], antioxidant vitamins, synthetic and natural antioxidants, plant-derived phenolics, phytochemicals, or fish lipids could prevent atherogenesis in animal models but were found to be inefficient in human clinical trials. N-acetyl cysteine (NAC), a precursor of GSH synthesis, may neutralize RCS via its antioxidant, anti-inflammatory, and RCS-scavenger properties [242]. NAC decreases atherosclerosis in animal models, but clinical studies in CVD patients provided limited or inconsistent results [242]. In addition, antioxidants are unable to neutralize RCS bioreactivity once they are formed [11].

Carbonyl scavengers could react and neutralize RCS to prevent protein lipoxidation. As reported by Colzani et al. [243], their efficacy and reactivity differ as function of RCS, with carnosine being the most efficient towards HNE, pyridoxamine towards MDA, aminoguanidine towards methylglyoxal and glyoxal, and hydralazine towards all RCS. Carnosine (β-alanyl-L-histidine) is a dipeptide available in food supplementation. It exhibits mild antioxidant, high carbonyl-scavenger, anti-glycating, and anti-inflammatory properties [244]. Carnosine prevents the development of lesions in animal models for atherosclerosis [245], but little is known about its efficacy in humans. As L-carnosine is rapidly inactivated by carnosinases in humans [246], carnosinase-resistant derivatives of carnosine (D-Carnosine octylester) have been synthesized and have shown high efficacy in preventing lipoxidation in vascular cells and atherosclerosis in animal models [247], thereby confirming the high therapeutic potential of carnosine for vascular diseases [244,248]. Hydralazine is an antihypertensive drug with antioxidant, metal-chelator, and aldehyde-scavenger properties, particularly on HNE and acrolein [249]. Hydralazine reduces the development of lesions in hypercholesterolemic rabbits and apoE^−/−^ mice [250]. No studies have been carried out on human patients so far.

Though promising results could be expected from carbonyl scavengers, the current challenge concerns their adequate bioavailability (which is very low) and their possible toxicity at efficient concentrations [251].

## 5. Conclusions

LDL oxidation in the intima generates multiple lipid oxidation products, including the highly bioreactive short-chain RCS, which form adducts on cysteine, lysine, and histidine epitopes on proteins, resulting in the formation of advanced lipoxidation end products, e.g. protein lipoxidation. This process impacts major systems and functions in the vascular wall, including the formation of foam cells via the uptake of oxLDLs by the scavenger receptor system present on macrophages and vascular cells, as well as endothelium dysfunction, cytoskeletal rearrangement, protein degradation, and ferroptosis. The real implication of lipoxidation throughout atherosclerosis is still not elucidated and has probably been underestimated, due (in part) to the poor efficacy of antioxidants in mitigating atherosclerosis complications. The extent of protein lipoxidation depends on the intensity of oxidative stress and exposure to RCS, with low levels of lipoxidation possibly involved in cellular defenses, while high levels could be associated with an impairment of tissular and cellular homeostasis. In advanced lesions, RCS accumulation (and subsequent lipoxidation) could contribute to plaque fragilization and rupture. Lipoxidation represents a challenging therapeutic strategy in atherosclerosis, justifying carrying out deeper investigations for a better understanding of this process, i.e adduct formation, identification of targets and evaluation of the functional consequences in the vascular wall.

## Figures and Tables

**Figure 1 antioxidants-13-00232-f001:**
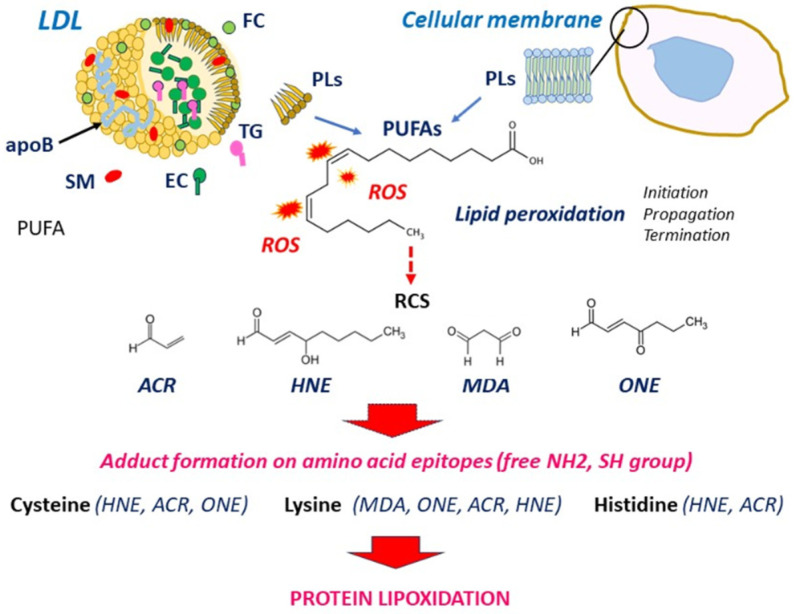
Formation of RCS from PUFA peroxidation in LDLs and cellular membranes. ROS attack PUFAs in LDLs and cellular membranes, which generates lipid oxidation products including RCS. RCS bind and form adducts on free amino groups (lysine, histidine) and thiol groups (cysteine), leading to protein lipoxidation. EC, esterified cholesterol; FC, free cholesterol; PLs, phospholipids; ROS, reactive oxygen species; SM, sphingomyelin; TG, triglycerides.

**Figure 2 antioxidants-13-00232-f002:**
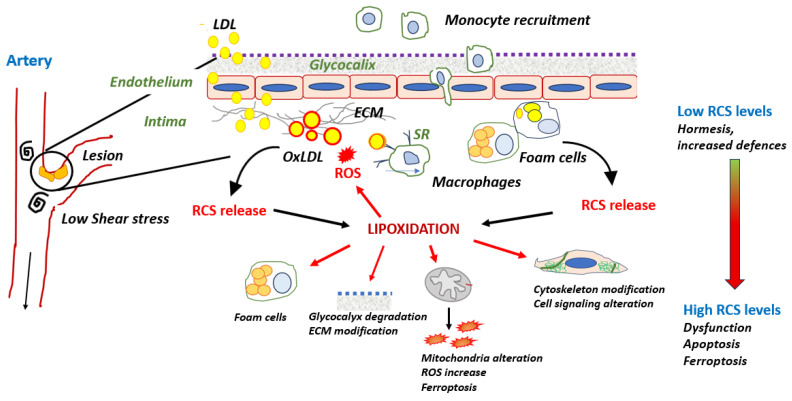
Cellular systems affected by lipoxidation and possibly detected in the intima. RCS-modified apoB promotes the uptake of oxLDLs by macrophages via SR receptors and the formation of foam cells. OxLDLs release RCS in the intimal environment or from macrophagic foam cells. RCS bind various cellular protein systems, thereby promoting their lipoxidation.

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
