# Peer review of "Reactive Carbonyl Species and Protein Lipoxidation in Atherogenesis"

_antioxidants, 2024, doi:10.3390/antiox13020232_

Round 1
Reviewer 1 Report
Comments and Suggestions for Authors
antioxidants-2825257
General comments
In this review, the authors discussed the multifactorial nature of atherosclerosis, a disease characterized by the accumulation of lipid-rich plaques in arteries, contributing significantly to cardiovascular diseases and global mortality. The authors highlight the potential crucial role of redox imbalance and lipid peroxidation in the development of atherosclerosis, leading to various responses such as endothelial activation, inflammation, and foam cell formation. The paper emphasizes protein lipoxidation, a non-enzymatic post-translational modification induced by reactive carbonyl species (RCS) on target proteins. The modification of low-density lipoproteins (LDL) and apolipoprotein B (apoB) by RCS is identified as a major contributor to foam cell formation, a key process in atherosclerosis. Additionally, oxidized LDL serves as a source of RCS, forming adducts on numerous proteins, impacting various systems, including extracellular matrix components, membranes, cytoplasmic and cytoskeletal proteins, transcription factors, and other cellular components. Despite the acknowledged significance of protein lipoxidation in atherogenesis, the mechanisms underlying lipoxidation-induced vascular dysfunction remain incompletely understood. The review primarily focuses on exploring protein lipoxidation during the progression of atherosclerosis.
Special comments:
The authors reviewed numerous references published 15-20 years ago. It would have a more significant impact on the field if the authors could incorporate more recent research findings into their review.
Author Response
We thank the reviewer for his/her encouraging comments.
Special comments:
The authors reviewed numerous references published 15-20 years ago. It would have a more significant impact on the field if the authors could incorporate more recent research findings into their review.
We agree with the reviewer’s suggestion which has been taken into account. In the new version of the manuscript, we have incorporated more recent research and review articles, so that less than 30 % of the quoted references are from before 2010. However many original articles describing RCS-adduct formation on protein systems, were published between 1990 and 2010, and are still relevant. It would be therefore difficult not to mention them.
Reviewer 2 Report
Comments and Suggestions for Authors
The authors reviewed the role of reactive carbonyl species (RCS) and protein lipoxidation in atherosclerosis, a chronic disease of arteries that causes cardiovascular complications. The paper briefly describes the main stages of atherosclerosis, from endothelial dysfunction and LDL oxidation to plaque formation and rupture, highlighting the involvement of oxidative stress and inflammation. The paper focuses on the effects of RCS, such as HNE, MDA, ACR, and ONE, that are generated by lipid peroxidation of PUFAs in LDL. RCS covalently modify proteins, forming advanced lipoxidation end-products (ALEs), and affect various cellular and extracellular systems, such as inflammation, ROS production, endothelial barrier function, and foam cell formation. Specific comments:
1. The section 2.1 on endothelial dysfunction is clear and comprehensive. However, it could be more focused on the role of RCS in endothelial dysfunction, rather than describing the general mechanisms of oxidative stress and NO bioavailability. For example, the authors could mention how RCS affect the expression and activity of eNOS, or how RCS interact with NO and other ROS to form peroxynitrite and other reactive species.
2. The section 2.2 on LDL transcytosis is interesting and relevant. However, it could be more connected to the main topic of the paper, which is protein lipoxidation. For example, the authors could discuss how LDL transcytosis affects the exposure of LDL to oxidative stress and the formation of RCS, or how LDL transcytosis is regulated by lipoxidation products or modified proteins.
3. The section 2.3 on LDL oxidation and foam cell formation is well-organized and informative. However, it could be more balanced and include some recent findings and controversies. For example, the authors could mention the role of myeloperoxidase and other enzymes in LDL oxidation, or the heterogeneity and plasticity of macrophages and foam cells in atherosclerosis.
4. The section 2.4 on advanced lesions is concise and accurate. However, it could be more updated and include some novel concepts and mechanisms. For example, the authors could mention the role of autophagy, ferroptosis, and extracellular vesicles in plaque progression and rupture, or the influence of microbiota and metabolites on plaque inflammation and stability.
5. The section 3 on RCS in early atherosclerosis lesions is the core of the paper and provides a comprehensive and detailed review of the literature. However, it could be more structured and coherent, and avoid overlapping or redundant information. For example, the authors could organize the section into subsections based on the different types of RCS (e.g., HNE, ONE, ACR, MDA, etc.), or based on the different targets and effects of RCS (e.g., lipoproteins, proteins, signaling pathways, etc.).
6. The section 4 on RCS in advanced atherosclerosis lesions is also important and relevant. However, it could be more consistent and clear, and avoid mixing different levels of analysis. For example, the authors could separate the discussion on the effects of RCS on cellular functions (e.g., apoptosis, senescence, proliferation, etc.) from the discussion on the effects of RCS on plaque characteristics (e.g., necrotic core, fibrous cap, calcification, etc.).
7. The section 5 on therapeutic strategies is brief and speculative. However, it could be more comprehensive and critical, and include some evidence and challenges. For example, the authors could review some existing or potential drugs or natural compounds that target RCS or lipoxidation, or discuss some limitations or side effects of these interventions.
8. The quality (resolution) of Table I is not good enough.
Author Response
We thank the reviewer for her/his encouraging comments. As suggested, the structure of the review has been modified. The part related to advanced lesions has been improved particularly missing points that have been developped (microcalcifications). A pharmacological section (section 4) has been included. Table I was moved in the first part of section 3 (RCS) and the quality is (hopefully) better. All changes are surlined in yellow.
Points 1,2,3
The section 2.1 on endothelial dysfunction is clear and comprehensive. However, it could be more focused on the role of RCS in endothelial dysfunction, rather than describing the general mechanisms of oxidative stress and NO bioavailability… The section 2.2 on LDL transcytosis could be more connected to the main topic of the paper, which is protein lipoxidation.
The suggestion of developping RCS in section 2 is not easy. Section 2 is a brief overview on the pathophysiology of atherosclerosis, mainly focused on the early steps of atherogenesis, and the role of RCS is described subsequently in section 3. In order to clarify this point, we introduced a sentence at the end of the introduction to Section 3. As suggested by the reviewer, the Table I has been moved in this Section 3.
The section 2.3 on LDL oxidation and foam cell formation is well-organized and informative. However, it could be more balanced and include some recent findings and controversies.
The different systems involved in LDL oxidation are briefly mentioned, including myeloperoxidase and other enzymes. However this highly investigated and complex topic was not further developped in this review as the main subject concerns lipoxidation in the pathophysiology of atherosclerosis.
Point 4
The section 2.4 on advanced lesions is concise and accurate. However, it could be more updated and include some novel concepts and mechanisms...
As suggested by the reviewer, we improved the section based on the pathophysiology of advanced lesions with focus on autophagy, ferroptosis and calcifications.
Point 5
The section 3 on RCS in early atherosclerosis lesions is the core of the paper and provides a comprehensive and detailed review of the literature. However, it could be more structured and coherent, and avoid overlapping or redundant information. For example, the authors could organize the section into subsections based on the different types of RCS (e.g., HNE, ONE, ACR, MDA, etc.), or based on the different targets and effects of RCS (e.g., lipoproteins, proteins, signaling pathways, etc.).
It is difficult to present the data with subsections based on the different types of RCS, because most studies have been done with HNE and at a lesser extent with MDA. Regarding the reviewer’s suggestion (e.g. presenting the effects of RCS on the different protein targets), the review was organized this way.
Point 6
…, the authors could separate the discussion on the effects of RCS on cellular functions (e.g., apoptosis, senescence, proliferation, etc.) from the discussion on the effects of RCS on plaque characteristics (e.g., necrotic core, fibrous cap, calcification, etc.).
We have introduced a new section 3.3. with the implication of lipoxidation in advanced lesions. The effects of RCS on the different items (apoptosis, senescence, calcification…) are discussed separately as suggested.
Point 7
The section 5 on therapeutic strategies is brief and speculative. However, it could be more comprehensive and critical, and include some evidence and challenges...
We have introduced a new section on pharmacological interventions aiming at preventing and neutralizing lipoxidation. So far, most studies were based on antioxidant therapies giving desappointing results.
Point 8.
The quality (resolution) of Table I is not good enough.
We tried to improve the quality of this Table. We hope that the resolution is better in the new version of the manuscript.
Reviewer 3 Report
Comments and Suggestions for Authors
The authors of this review article summarized current knowledge on the biology of reactive carbonyl species (RCS) and their role in the oxidation of lipoproteins (rendering them dysfunctional and prone to atherosclerosis promotion) with a special emphasis on protein lipoxidation and related vascular dysfunction. The review contains a large amount of information and addresses numerous molecular and cellular processes set into operation by RCS and linked to atherosclerosis. The material is scientifically sound and the review addresses an important problem. I have a few comments that may need addressing by the authors:
The large amount of information and numerous mechanisms and cellular processes through which RCS damage vascular structures and promote atherosclerosis makes the material difficult to follow. Thus, I advise the authors to improve the structure of the material to improve readability. In the view of this reviewer, the review benefits from a scheme to summarize the chemical structure of RCS, their substrates (sources), mechanism of synthesis and mechanisms via which they exert their deleterious effects. The authors may emphasize that low levels of RCS may be not deleterious because they participate in physiological cellular signaling and thus there is a dose-effect relationship in their biological actions. The use of a large amount of abbreviations adds further difficulties in following the material. If the authors could use less abbreviations (particularly amino acid symbols), the readability could be improved.
The participation of RCS in initiation, build-up and destabilization of atherosclerotic plaques needs to be clearer presented. If the material could be structured this way, the clinical value of the review could have been improved. In general, the review should be made more clinically oriented.
Are any pharmacological (or other ways, like diet or life style modifications) to reduce the production and impact of RCS and their effects.
The participation of RCS in vascular (and plaque) calcification was not mentioned. Vascular calcification at the sites of atherosclerotic plaques is common.
There are many awkward sentences throughout the material. Many sentences, particularly long and complex sentences with multiple notions may be split into smaller ones.
The authors should be uniform in using references, particularly when they name the authors. The correct form of referencing is Author et al.(reference) and not Author et al. and the reference placed at the end of sentence.
The authors placed a table within the conclusions of the study. I advise the authors to place the table before the conclusions. In addition, all abbreviations should be explained in full wording in the footnote to table.
Can the authors other some implications of the review and future dirrections?
Comments on the Quality of English LanguageModerate editing of English language required
Author Response
We thank the reviewer for her/his encouraging comments. As requested, we provide a point per point response to the suggestions. The part related to advanced lesions has been improved particularly missing points that have been developped (microcalcifications). A pharmacological section (section 4) has been included. Table I was moved in the first part of section 3 (RCSand a new Figure 1 has been included with some RCS formulae and function. All changes are surlined in yellow.
Point 1. Scheme to summarize the chemical structure of RCS, their substrates (sources), mechanism of synthesis and mechanisms via which they exert their deleterious effects
As suggested, we prepared a new Figure 1 presenting the chemical structure of RCS and their origin from PUFA peroxidation. The deleterious effects are briefly presented in Figure 2 and summarized in Table I. Note that the structure, synthesis and reactivity of RCS have been largely described and illustrated, including in the cardiovascular system (ref.12) and the references are quoted in our manuscript (see references 12, 14, 17-20, 62-66…).
Point 2. The authors may emphasize that low levels of RCS may be not deleterious because they participate in physiological cellular signaling and thus there is a dose-effect relationship in their biological actions.
We thank the reviewer for suggesting to improve this part of the manuscript. In the new version, we introduced a new paragraph (top section 3), describing this biphasic aspect of RCS properties.
Point 3. The use of a large amount of abbreviations adds further difficulties in following the material. If the authors could use less abbreviations (particularly amino acid symbols), the readability could be improved.
We suppressed amino acid abbreviations in the full text, and replaced them by the full name. Abbreviations were kept in Table I and are explained in the legend. The list of abbreviations on footnotes in the first page of the manuscript has been completed.
Point 4. The participation of RCS in initiation, build-up and destabilization of atherosclerotic plaques needs to be clearer presented. If the material could be structured this way, the clinical value of the review could have been improved. In general, the review should be made more clinically oriented.
In our manuscript, we present the role of RCS in the formation of early lesions, with first the formation of foam cells, inflammation and endothelial dysfunction. As suggested, the role of RCS in advanced lesions has been partly developped in the new version of the manuscript.
Are any pharmacological (or other ways, like diet or life style modifications) to reduce the production and impact of RCS and their effects.
We have introduced a new section 4 on pharmacological interventions aiming at preventing and neutralizing lipoxidation. So far, most studies are based on antioxidant therapies giving desappointing results.
The participation of RCS in vascular (and plaque) calcification was not mentioned. Vascular calcification at the sites of atherosclerotic plaques is common.
As suggested, we added a new paragraph describing the implication of RCS in microcalcifications. Note that lipoxidation targets are not known on this subject.
There are many awkward sentences throughout the material. Many sentences, particularly long and complex sentences with multiple notions may be split into smaller ones.
The text has been carefully proofread and some sentences have been shortened and simplified, in order to clarify the presentation of the review.
The authors should be uniform in using references, particularly when they name the authors. The correct form of referencing is Author et al.(reference) and not Author et al. and the reference placed at the end of sentence.
This has been corrected accordingly
The authors placed a table within the conclusions of the study. I advise the authors to place the table before the conclusions. In addition, all abbreviations should be explained in full wording in the footnote to table.
Table I has been moved to the top of section 3 (focused of the effect of RCS). Most abbreviations have been explained accordingly